# Body Balance Analysis in the Visually Impaired Individuals Aged 18–24 Years

**DOI:** 10.3390/ijerph192114383

**Published:** 2022-11-03

**Authors:** Marta Urbaniak-Olejnik, Wawrzyniec Loba, Olgierd Stieler, Dariusz Komar, Anna Majewska, Anna Marcinkowska-Gapińska, Dorota Hojan-Jezierska

**Affiliations:** 1Department of Hearing Healthcare Profession, Poznan University of Medical Sciences, 61-701 Poznań, Poland; 2Department of Biophysics, Poznan University of Medical Sciences, 61-701 Poznań, Poland

**Keywords:** postural stability, visually impaired, visual dysfunction, balance

## Abstract

Background: Proper body balance is maintained by both sensory, i.e., vision, hearing, vestibular, and proprioception, and motor nervous systems. Visual control facilitates balance both in a static position, as well as during movement. Visual dysfunctions can adversely affect balance and stability control. Methods: The study comprised a group of 30 individuals visually impaired from birth aged 18–24 years. The control group consisted of 50 persons in the same age range as the study group. The trial was performed as four separate tests: two on a stable surface platform, and two on an unstable surface platform. Results: No statistically significant differences were found in the posturography test on the stable platform between the study groups. When tested on an unstable ground surface, the visually impaired subjects obtained showed worse performance than the sighted persons. Statistically significant differences were observed in the majority of the analyzed results. Conclusions: Visually impaired individuals present a poorer ability to maintain balance under dynamic conditions. Tests conducted on the unstable surface platform seem to be more relevant and applicable in the diagnosis of the subjects with visual impairment, as they differentiate the subjects to a greater extent.

## 1. Introduction

The primary function of the balance system is to provide a stable body posture, which in turn allows efficient movement. Furthermore, the proper functioning and interaction of the visual, vestibular, and proprioceptive sensory organs, which are coordinated by the central nervous system, are essential for maintaining body balance [1,2,3]. By means of eye stabilization, the visual system controls the direction and focus of vision during head and body movements in relation to the surrounding objects. The vestibular system indicates the linear and angular accelerations experienced by the head due to the impact of external forces, such as gravity. Moreover, the proprioceptive sensory receptors provide information regarding the degree of muscle tone, the joint position, and the force of foot pressure on the ground [4,5].

There are a number of scientific reports investigating the process of maintaining balance in non-disabled individuals with and without full visual control. The results of these studies clearly demonstrate that the range of sways in the stabilogram is increased when subjects are tested with eyes closed, implying a deterioration in postural stability. Lack of visual control results in reduced angular and linear position information and insufficient visual stimulus input leading to a regression in stability [6,7,8]. The formation of postural movement coordination among the visually impaired is still insufficiently understood. Furthermore, the available literature contains inconclusive findings regarding the effect of vision loss on balance performance. Some studies have shown that individuals with visual impairment are able to maintain balance better than persons without visual dysfunctions [9], whereas other studies have indicated the opposite results [10,11,12,13]. The ability to control the body is vital in the development of motor skills, physical awareness, and spatial orientation, which is crucial during the rehabilitation process for persons suffering from visual impairment [14].

Posturography is a commonly used diagnostic tool that allows for the quantitative assessment of balance system function. The test records and analyzes the displacement of the center of gravity projection with respect to the surface. During a single trial, a series of tests are performed with the eyes open and closed, which can be performed on both stable and unstable ground surfaces.

The purpose of this study is to evaluate body balance in individuals with a visual impairment from birth, between the ages of 18 and 24. The study poses the following research questions:Are there differences in the body balance of the visually impaired and sighted persons on a stable and unstable platform surface?Is the body balance level of the sighted individuals reduced in comparison to the visually impaired persons when visual perception is disengaged?

## 2. Materials and Methods

Participation in the study was entirely voluntary, and volunteers were free to withdraw from the study at any time. In addition, before the onset of the study, each volunteer was presented with the study’s methodology, as well as informed of the study’s purpose, its detailed course, and duration. All subjects had to provide informed consent. Prior to the study, a brief interview was conducted with the subjects, aimed at identifying potential balance issues, the presence of vertigo, excluding orthopedic issues, as well as at coping with everyday situations. The patency of the ear canals was also verified and the condition of the eardrum was assessed; additionally, body height and weight were also evaluated. Body height was measured to an accuracy of 0.5 cm, whereas body weight was measured to an accuracy of 10 g. Body weight measurement was performed with no footwear and in light clothing. These parameters were collected in order to calculate BMI (Body Mass Index). Before the study, hearing thresholds for the four frequencies of 500 Hz, 1000 Hz, 2000 Hz, and 4000 Hz were also tested using Madsen Itera 2 audiometer (GN Otometrics A/S, Taastrup, Denmark) equipped with TDH-39 air-conduction headphones. The test was performed in a dedicated sound-attenuated room. The test was performed with the 2/3 ascending method for both ears, starting with the ear that the subject indicated as having slightly better hearing. The stimulus was applied continuously by a 5 dB HL step size.

Fifty subjects who were visually impaired from birth took part in the study. Since the exclusion criteria were any vestibular abnormalities or orthopedic issues, the results of 30 participants constituted the analysis. The group included 10 men and 20 women in the age range of 18–24 years, with an average age of 19.7 years. Additionally, the study included volunteers presenting normal intellectual functioning, as determined by a certificate issued by a psychological and pedagogical counseling center, who were ontologically healthy. The study employed the WHO (1992) classification, according to which the visually impaired have a visual acuity of less than 3/60 Sn and/or a limited visual field of up to 20 degrees. The study group included students of the Special Education Centre for Blind Children in Owińska. 

The control group consisted of 50 individuals without visual problems (no use of corrective glasses or contact lenses), 32 women and 18 men aged between 18 and 24, with an average age of 20.9 years. Detailed data characterizing the groups are presented in Table 1.

Evaluation of balance performance was conducted using the MediBalance Pro system (MediTECH Electronic GmbH, Wedemark, Germany) consisting of a stabilometric platform and a computer program for assessing the functional state of the balance system (Figure 1). The standard parameters determined via posturography were analyzed: mean sway radius (mm), mean speed (cm/s), and area (cm^2^).

The subjects completed four tests in a standing position with eyes open and closed on a stable and unstable platform surface. A foam cushion provided by the manufacturer was used to simulate the unstable ground. Each test on the stability platform lasted 30 s with 15 s breaks. The entire test was conducted during a single appointment. Among the visually impaired, only the individuals who possessed eyeball prostheses (3 persons) did not perform closed-eye trials. The remaining subjects who presented with a sense of light or a residual vision performed all trials. 

Statistical analysis was performed for the obtained findings. The conformity of the parameters to a normal distribution was tested using the Shapiro–Wilk test, including the Kolmogorov–Smirnov test with Lilliefors correction, (*p* > 0.05). The arithmetic mean and standard deviation were calculated for each parameter. Analyses of the significance of statistical differences between the compared groups of people with and without visual impairment were performed. Due to the absence of a normal distribution, the ANOVA, Kruskal–Wallis test was performed. For dependent samples’ comparison of parameters of individual tests in each group, the Anova Friedman test was used. Dunn’s post hoc test was also performed. Hypothesis verification was tested at the two-sided statistical significance level of α ≤ 0.05. 

The Bioethics Committee’s resolution, dated 1 February 2018–Resolution No. 121/18, approved the implementation of the abovementioned study.

## 3. Results

The analyses of the body structure of the visually impaired and sighted individuals aged 18–24 years indicate that the body proportions of both men and women in these groups were similar. Among the sighted men, as well as women, both greater body weight and height were observed, although no statistically significant differences were found between the parameters. Detailed data are presented in the table below (Table 2).

According to the analyzed posturographic tests performed on the stable surface platform, there were no statistically significant differences between the study groups. However, it was found that when the visual control was disengaged, the results of the parameters decreased in both the visually impaired and sighted subjects, and these results were statistically significant. In addition, comparisons of the stabilograms of individuals with and without visual impairment demonstrated that the persons with the normal vision presented higher values in the analyzed parameters than the visually impaired. This, in turn, may indicate poorer body coordination in the stable platform test. Detailed data are presented in the following table (Table 3).

Comparing the body balance parameters of the visually impaired and sighted subjects on the unstable surface platform, a distinct increase in all parameter values is observed for both open and closed eyes. In the test on the unstable surface platform, individuals with visual impairment obtained higher mean values than individuals with normal vision. In addition, statistically significant differences were found in most of the analyzed results. Detailed data are presented in Table 4. These results may indicate that visually impaired persons present a poorer ability to maintain balance in dynamic conditions.

## 4. Discussion

Balance is the ability that allows the body to maintain its spatial position. Additionally, a prerequisite for a stable posture is sustaining the center of gravity projection within the foot support surfaces. However, this ability tends to be an individual trait that is genetically and environmentally determined [15].

In fact, the vestibular apparatus, the vision, and the proprioceptive sensory organ are involved in determining the position of the center of gravity in relation to the gravitational forces. Nevertheless, their functionality varies depending on external conditions [3,16] The function of the balance system is to provide information regarding the position, the speed of body movement, and the deflection of the center of gravity, as well as to monitor the eye movement [17].

Within the system maintaining balance, it is possible to distinguish two distinct systems which are mutually dependent, i.e., the somatosensory system, which stabilizes the eye, and the vestibular system, which stabilizes posture both at rest and in movement. The function of the somatosensory system is to control vision direction and focus during movements of the head and body. In turn, the vestibular system transmits information about the position of the head in relation to forces acting on it, such as gravity, or linear and angular accelerations. Additionally, the vestibular system affects skeletal muscle function by means of vestibulospinal reflexes [5,18]. However, in order for the balance system to function properly, symmetry in terms of receiving and transmitting stimuli from both vestibules is essential. In fact, disturbance of either system may result in difficulties with functioning and movement, as well as integration in space [19]. According to the literature, visual dysfunctions may reduce the amount of information regarding the body’s spatial position, which ultimately negatively affects the stability of the body, generating greater imbalance and increasing the risk of falls [7,20].

In the control group under stable conditions, statistically significant differences were observed between measurements with eyes open and closed for speed and area parameters. Despite an increase in the mean for sway, no statistically significant difference was observed. Under unstable conditions, differences were observed for all measured parameters. The effect of closing the eyes on the deterioration of stabilometric parameters in a similar age group is also shown by Adamo et al. [21]. In Adamo et al. paper, the differences in sway are statistically significant in both conditions (stable and unstable), but much more distinct differences in the averages are observed in the unstable conditions [21].

In the present study, a group of the visually impaired aged 18–24 years obtained slightly poorer results in body coordination in an upright position with the eyes open on the stable surface platform than the controls, although the results were not statistically significant. In the case of tests conducted with the eyes closed, no evident increase in the analyzed parameters was observed among the visually impaired subjects as compared with the tests with eyes open; however, the analyses indicated that the results were statistically significant. A comparison of the stabilograms for the analyzed groups also revealed that the sighted individuals obtain poorer scores when visual control is disengaged in comparison with the visually impaired, which in turn reflects the poorer efficiency of body coordination. The results of the stabilograms on an unstable surface are entirely different. The visually impaired present a poorer ability to maintain body balance in dynamic conditions, as confirmed by the analyses demonstrated above.

Similar results to our own study were obtained by Ray et al., who conducted their study on a group of 23 individuals with visual impairment. According to their analysis, there were no statistically significant differences between the sighted subjects and the studied group on the stable surface platform in measuring conditions. In contrast, a reduction in postural stability was observed in dynamically changing conditions (unstable ground) [22]. Furthermore, in a study by Alighadir et al., the visually impaired subjects obtained poorer stabilogram values on a foam surface than the subjects with normal vision [23]. Similarly, a study by Wiszomirska et al. conducted on a group of women with visual dysfunction revealed significantly worse results on an unstable surface compared with sighted women [24]. Moreover, Weirech et al. also found that the greatest differences occurred on unstable surfaces [25].

In the study by Giagazoglou, the visually impaired exhibited slightly deteriorated balance level compared with the sighted individuals. Following the analysis, it was concluded that the balance of subjects with normal vision deteriorates significantly when visual control is disengaged, which was not observed in individuals with visual impairment [8]. 

According to the literature, maintaining even a limited visual field significantly facilitates spatial orientation, therefore, visually impaired individuals compensate for imbalances more quickly and easily. In addition, certain studies have demonstrated that persons suffering from visual impairment are able to maintain better balance during static or dynamic postural tasks, in comparison to individuals with normal vision, although other studies have established the opposite relationship. The results of the study by Alghardir et al. and Schmid et al. confirm that vision-impaired subjects, regardless of whether their eyes are open or closed, behave in the same way as sighted individuals with their eyes closed [18,23]. In our study, in the group of visually impaired patients, for some of the parameters (sway and area on a stable surface, sway on an unstable surface) no significant differences between results with opened and closed eyes were observed. Moreover, even if a statistically significant difference occurred, values of the parameters measured with eyes closed were close to the ones measured with eyes opened, and for some parameters, they showed better results than the control group. Thus, our results are consistent with the aforementioned studies by Alghardir et al. and Schmid et al. observations.

Despite numerous reports on balance studies in hearing-impaired individuals, none of the papers cited in this article focused on comparing stabilometric parameters in the 18–24 age group. Above that, most of the studies are done on a dynamic platform while our study presents results on an unstable platform–despite this, the findings are comparable. Body balance assessment is extremely difficult, due to the involvement of multiple mechanisms affecting its control. According to the literature analysis, the greatest involvement in the balance control system is observed in the dynamic tests, which is also confirmed by the presented research. Hence, posturographic studies on the unstable surface platform reflect the complexity of maintaining the balance when the mechanism is affected by internal and external disturbances.

## 5. Conclusions

On the basis of the analyzed results of the presented study, it can be concluded that there is not enough evidence to determine the relationship between the level of visual dysfunction and static balance. Nevertheless, a significant deterioration in the ability to maintain balance in dynamic conditions is evident. Future research should focus on dynamic conditions and situations which additionally impede focusing on postural control by increasing the demands associated with cognitive task processing.

The conducted balance analyses among the visually impaired subjects provide important insight into the development of this ability to maintain proper body control. However, definite conclusions based on all the findings obtained should be formulated with caution.

## Figures and Tables

**Figure 1 ijerph-19-14383-f001:**
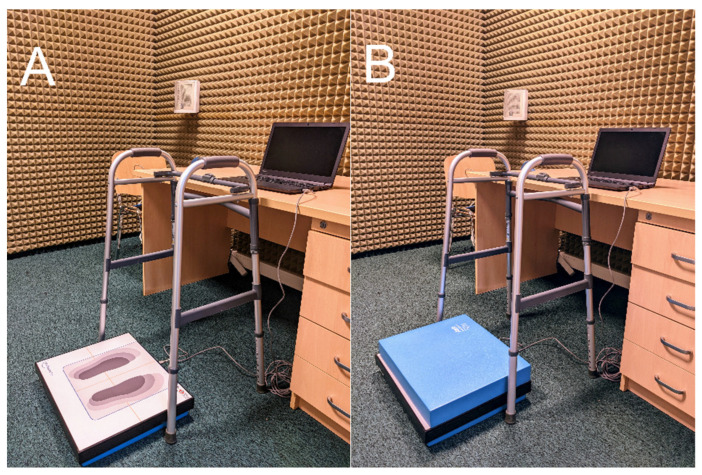
Photo of a setup of MediBalance Pro system: (**A**)—stable setup; (**B**)—unstable setup.

**Table 1 ijerph-19-14383-t001:** Characteristics of the analyzed groups with respect to group size and subjects’ age.

Subjects	Study Group		Control Group	
Gender	Female	Male	Female	Male
Number	20	10	32	18
Age	19.5	20.1	20.9	21.0
Standard deviation	(1.7)	(2.1)	(2.0)	2.2

**Table 2 ijerph-19-14383-t002:** Body parameters in the study and control groups.

Subjects	Study Group		Control Group	
Gender	Female	Male	Female	Male
Body height (cm)	165.4 ± 2.4	174.6 ± 2.7	167.9 ± 2.9	178.0 ± 3.0
Body mass (kg)	52.3 ± 2.7	68.5 ± 2.3	54.7 ± 3.1	72.2 ± 2.8
BMI	19.4 ± 2.3	22.2 ± 2.6	19.9 ± 2.8	22.9 ± 2.4

**Table 3 ijerph-19-14383-t003:** A comparison of balance parameters in the study and control group on the stable surface platform.

Stable Surface Platform	Analyzed Groups							
Balance Parameters	Study Group			*p*-Value Eyes Open between Groups	*p*-Value Eyes Closed between Groups	Control Group		
	Eyes Open	Eyes Closed	*p*-Value in the Study Group			Eyes Open	Eyes Closed	*p*-Value in the Control Group
Sway (mm)	3.7 ± 1.9	4.4 ± 1.7	0.51	0.81	<0.05 *	3.5 ± 2.3	4.5 ± 2.0	0.37
Speed (cm/s)	5.5 ± 2.1	6.3 ± 2.7	<0.05 *	0.79	<0.05 *	5.4 ± 3.3	8.4 ± 3.1	<0.05 *
Area (cm^2^)	1.1 ± 0.7	1.2 ± 0.9	0.58	0.88	<0.05 *	1.1 ± 0.9	1.9 ± 1.0	<0.05 *

* Statistically significant difference.

**Table 4 ijerph-19-14383-t004:** A comparison of balance parameters on the unstable surface platform in the study and control groups.

Unstable Surface Platform	Analyzed Groups							
Balance Parameters	Study Group			*p*-Value Eyes Open between Groups	*p*-Value Eyes Closed between Groups	Control Group		
	Eyes Open	Eyes Closed	*p*-Value in the Study Group			Eyes Open	Eyes Closed	*p*-Value in the Control Group
Sway (mm)	7.4 ± 2.6	7.9 ± 2.7	0.88	<0.01 *	<0.01 *	5.7 ± 1.9	7.3 ± 2.8	<0.01 *
Speed (cm/s)	15.2 ± 3.7	16.4 ± 2.7	<0.05 *	<0.01 *	<0.05 *	10.1 ± 1.6	16.0 ± 5.2	<0.05 *
Area (cm^2^)	4.7 ± 0.4	5.0 ± 2.2	<0.01 *	<0.05 *	<0.05 *	3.2 ± 1.3	6.3 ± 2.4	<0.05 *

* Statistically significant difference.

## Data Availability

Not applicable.

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
