# Peer review of "Body Balance Analysis in the Visually Impaired Individuals Aged 18–24 Years"

_ijerph, 2022, doi:10.3390/ijerph192114383_

Round 1
Reviewer 1 Report
This study investigates the impact of visual control on body balance in sighted and visually impaired participants. They found that the performance was significantly different between these two groups in the eye-closed conditions on stable surfaces and in the eye-open and eye-closed conditions on unstable surfaces. The results are clear and support the conclusions. The findings shed new light on the understanding of the role of vision in human posture control. I only have several minor points.
- In the abstract, "When tested on the unstable ground surface, the visually impaired subjects obtained higher mean values than the sighted persons." "higher mean values" are not apparent to the audience at this stage. Change it to "showed worse performance"?
- It would be helpful to provide a figure to show the study's setup.
- Explain how to measure the "sways," "speed," and "areas" in detail.
- The t-test was performed multiple times in the analysis and must be corrected for multi-comparisons.
- Discussion "In the case of tests conducted with the eyes closed, no significant increase in the analyzed parameters was observed among the visually impaired subjects as compared to the tests with eyes open; however, the analyses indicated that the results were statistically significant." The word "significant" usually means "statistically significant". The authors may replace the first "significant" with "evident" or "apparent". In addition, if the difference is not evident but significant, the authors may need to check whether or not the statistical method is appropriate. As I suggested above, the correction for multiple comparisons is needed.
- In the Discussion, the authors listed several studies showing that similar findings have been reported before. Then what is new, and what can the current findings add to the literature?
Reviewer 2 Report
The authors stated that the purpose of the study was an investigation of body balance in a group of individuals with visual impairments since birth under two support surface conditions: stable and unstable. They were also interested if the latter surface condition affected the study group more than a sighted control group.
The manuscript is reasonably well-written and the study protocol does not have any obvious issues of concern. One may query why the study group was much smaller than the control group (30 versus 50). This discrepancy should be addressed in the manuscript. The main criticism of the manuscript is that the authors have not fully addressed the implications of the statistical results and so this needs to be thoroughly discussed before the manuscript should be considered for publication.
Comments and suggested edits:
Abstract: The first sentence is awkward in its phrasing and in the use of "organ of" (see below). Suggested re-write: "Proper body balance is maintained by both sensory, i.e., vision, hearing, vestibular, and proprioception, and motor nervous systems." Or, something similar.
Introduction: The use of the phrase "organ of" is unusual and it is probably better to say 'eyes' for 'organ of sight. (line 31)
Materials and Methods: For the control group, does 'without visual problems' mean all 50 individuals did not need any corrective lenses? If not, perhaps indicate how many did and what type of correction (near- versus far-sighted). (line 73)
Why did you recruit 50 control subjects while only 30 study subjects? Why not have the control sample equal the study sample? (line 73)
Results: Tables 3 and 4 need to be better formatted so that the headings do not dominate the tables.
Discussion: Delete "(vagus)" as it is confusing why you included that. Also, change "organ of sight" to "eyes" or "vision". (line 160)
Change "presented" to "present" (line 180)
Delete "conclusively" (line 225)
Issue to be addressed:
If one considers the balance data with a stable base of support as the baseline, there are clearly some issues with the control group as two of the three variables were significantly different within this group between eyes open and eyes closed. In fact, the absolute difference in the third variable was larger in the control group as compared to the study group. These data suggest that the control group was not homogeneous with respect to balance performance. Further exploration of the balance capabilities of this group will likely reveal possible issues with vision, vestibular function, and/or proprioception. Simply stating a conclusion that there is insufficient evidence in the data without explaining why the control group was not, in fact, a control group, or how the control group data may have influenced the interpretation of the study group data is not acceptable. Your analysis of the data needs to more in-depth and comprehensive.
